# Effect of Alumina and Silicon Carbide Nanoparticle-Infused Polymer Matrix on Mechanical Properties of Unidirectional Carbon Fiber-Reinforced Polymer

**S. M. Shahabaz** [1], **Pradeep Kumar Shetty** [1,*], **Nagaraja Shetty** [1,*], **Sathyashankara Sharma** [1], **S. Divakara Shetty** [2] **and Nithesh Naik** [1]

[1] Department of Mechanical & Industrial Engineering, Manipal Institute of Technology, Manipal Academy of Higher Education, Manipal 576104, India
[2] Mangalore Institute of Technology & Engineering (MITE), Badaga Mijar, Near Moodabidri, Mangalore 574225, India
* Correspondence: pradeep.shetty@manipal.edu (P.K.S.); nagaraj.shetty@manipal.edu (N.S.)

**Abstract:** Unidirectional carbon fiber-reinforced polymer nanocomposites were developed by adding alumina ($Al_2O_3$) and silicon carbide (SiC) nanoparticles using ultrasonication and magnetic stirring. The uniform nanoparticle dispersions were examined with a field-emission scanning electron microscope. The nano-phase matrix was then utilized to fabricate the hybrid carbon fiber-reinforced polymer nanocomposites by hand lay-up and compression molding. The weight fractions selected for $Al_2O_3$ and SiC nanoparticles were determined based on improvements in mechanical properties. Accordingly, the hybrid nanocomposites were fabricated at weight fractions of 1, 1.5, 1.75, and 2 wt.% for $Al_2O_3$. Likewise, the weight fractions selected for SiC were 1, 1.25, 1.5, and 2 wt.%. At 1.75 wt.% $Al_2O_3$ nanoparticle loading, the flexural strength modulus improved by 31.76% and 37.08%, respectively. Additionally, the interlaminar shear and impact strength enhanced by 40.95% and 47.51%, respectively. For SiC nanocomposites, improvements in flexural strength (12.79%) and flexural modulus (9.59%) were accomplished at 1.25 wt.% nanoparticle loading. Interlaminar shear strength was enhanced by 34.27%, and maximum impact strength was improved by 30.45%. Effective particle interactions with polymeric chains of epoxy, crack deflection, and crack arresting were the micromechanics accountable for enhancing the mechanical properties of nanocomposites.

**Keywords:** flexural property; interlaminar shear test; impact test; hybrid $Al_2O_3$ nanocomposites; hybrid SiC nanocomposites

## 1. Introduction

Carbon fiber-reinforced plastics (CFRP) have been widely used as an alternative to metals in various industrial structures, including aerospace, automotive, and marine, due to their high specific strength and stiffness [1–4]. Carbon fibers, amongst the other fiber composites (glass fiber, natural fiber), retain their tensile strength at very high temperatures and remain unaffected by the moisture conditions [5,6]. Additionally, carbon fibers have strong thermal and electrical conductivities and a low thermal expansion coefficient. These unique properties make them the most versatile material for manufacturing of various parts and main structures in aerospace, marine, and automobile sectors. Hence, maximum application of carbon fibers is found in the aerospace industry, as the structures are exposed to the various ranges of environments and temperature conditions [7,8].

In addition to different types of fiber-reinforced composites, continuous fiber-reinforced laminated composites are the most common alternatives to metallic parts, where the fibers provide superior in-plane performance. However, the lack of reinforcement in the thickness direction weakens the resistance to out-of-plane loading, making delamination more likely [9]. Several techniques for reinforcing laminates in the thickness direction have been

developed to address this issue, including textile reinforcements, interleaves, stitching, etc. These techniques, however, constantly reduce the in-plane properties. Mouritz et al. reported that the tensile strength of stitched composites was reduced to 55% of non-stitched composites due to defects introduced during the stitching process [9]. On the contrary, a prepreg system with fine polyamide particles dispersed on the surface was developed. Although the prepreg laminates show excellent resistance to microscopic damage caused under static [10,11] and fatigue loading [12,13], the concern remains with the increase in fabrication costs, as the prepreg system requires an autoclave system.

On the other hand, toughening the matrix resin could improve the interlaminar properties of CFRP. Numerous studies have been performed to improve the properties of epoxies by addition of fine particles such as boron nitride [14,15], molybdenum disulfide [16,17], boron carbide [18], multi-walled carbon nano-tubes [19–21], silica [22], nano-clay [23,24], graphene [25], alumina [26,27], silicon carbide [28], and carbon nanofibers [29,30]. Zhao et al. found improved epoxy resin fracture toughness by adding fine alumina ($Al_2O_3$) particles by 17% over neat epoxy resin [27].

The application of resin incorporated with fine nanoparticles and nanofibers has also been investigated to improve the mechanical properties of CFRP composites. Chisholm et al. reported that textile carbon fiber-reinforced epoxy composites with the addition of fine silicon carbide (SiC) particles accomplished improved bending, tensile, and fatigue properties [31]. Kadhim et al. discovered a maximum improvement in epoxy flexural strength at 4 wt.% $Al_2O_3$ [32]. De Souza and dos Reis used dynamic mechanical analysis (DMA) to show the cross-link density behavior between epoxy and $Al_2O_3$ nanoparticles concerning the volume fractions [33]. Srivastava et al. used fine graphite SiC to enhance the fracture toughness of carbon fiber-reinforced epoxy composites with SiC particles, increasing by 45% and 55% for modes I and II, respectively [34]. Khashaba et al., during their investigation, found maximum improvement in tensile strength and tensile modulus for 1.5 wt.% $Al_2O_3$ nanoparticles incorporated into Epocast 50-Al/946, used as an adhesive for bonded joints in composite structures [35].

Tensile and interfacial fracture toughness was evaluated by Su et al. by incorporating the micro- and nano-sized $Al_2O_3$ fillers into CFRP prepreg layers. They reported that the crack propagation was lower due to the presence of nanoparticles at the interface [36]. Kaybal et al. used an ultrasonic technique for dispersion of $Al_2O_3$ nanoparticles ranging from 1 to 5 wt.% in epoxy resin. The modified composite with 2 wt.% $Al_2O_3$ showed better impact damage resistance and higher energy absorption before failure [37]. Bazrgari et al. developed nanocomposites by dispersing $Al_2O_3$ nanoparticles in epoxy resin at 1 and 3 vol%. They found that at 3 vol%, the highest flexural strength and stiffness were obtained, whereas for 1 vol% $Al_2O_3$, the highest impact strength, lowest wear rate, and lowest coefficient of friction were noted [38].

From the previous studies, it is observed that with the increment of nanoparticles, the mechanical property increases. However, the mechanical property is quickly tarnished after an optimum loading level of nanoparticles. This is due to the interaction between nanoparticles and polymeric chains of the matrix being more vital than the particle–particle interaction; as a result, the mechanical properties improve [39]. However, with the increasing filler content, the reverse action causes the nanoparticle localization, resulting in lower mechanical properties [40]. The prolonged and high-speed mixing lowers the formation of the clustering of nanoparticles. The gaps are reduced, and thus there is a possibility of a high degree of interaction between the nanoparticle and polymer chains, enhancing the hydrogen bonding between the phases and improving mechanical and thermal properties [41]. In addition to the above approach, the other factor influencing the behavior of mechanical properties is the proper selection of the particle size. A smaller particle size and larger surface area result in better dispersion within macromolecules, improving the mechanical properties of nanocomposites [42–44].

Two such inorganic fillers where the researchers have shown interest in providing improved interfacial bond strength and enhancing mechanical properties by incorporating

into epoxy resin are nano $Al_2O_3$ and SiC. When uniformly dispersed into epoxy resin, these nanoparticles act as physical cross-links for the epoxy molecular chains. As per the authors' knowledge, although there are several experimental studies performed on incorporating these nanoparticles into epoxy resin and glass fiber-reinforced polymer (GFRP) composites for enhancing the mechanical properties, the experimental investigations by proper selection of the optimal loading range of these nanoparticles on improving the mechanical properties of unidirectional CFRP composites with the above-selected nanoparticles are significantly fewer. From the literature, it is also observed that there are limited research articles which provide the exact percentage weight ratio of the above nanoparticles incorporated into the polymer composite, to gain the maximum mechanical properties. Therefore, the authors in the present work have found the exact optimal weight fractions for the above nanoparticles in improving flexural, interlaminar shear, and impact properties by embedding them into a unidirectional carbon fabric. The mechanical properties obtained in the present experimental work will be further used in the machinability (drilling) studies of these composites in the future works.

## 2. Experimental Work

### 2.1. Materials

Fabrication of composites was performed using unidirectional woven carbon fabric T300-3K with 200 GSM. The polymer matrix bisphenol A diglycidyl ether (Part-A) with an amine-based hardener (Part-B) was used with the mixing ratio of 100:30, followed by room temperature curing. Fiber and resin properties are represented in Tables 1 and 2.

**Table 1.** Unidirectional carbon fiber properties.

| Property | Value |
|---|---|
| Density (g/cm$^3$) | 1.8 |
| Filament diameter (µm) | 7 |
| Tensile strength (MPa) | 4000 |
| Tensile modulus (GPa) | 240 |
| Elongation (%) | 1.7 |

**Table 2.** Typical properties of resin and hardener.

| Property | Test Method | Resin | Hardener |
|---|---|---|---|
| Viscosity at 25 °C (MPas) | ASTM D445 | 9000–12,000 | <50 |
| Density at 25 °C (g/cc) | ASTM D4052 | 1.2 | 0.95 |
| Flashpoint (°C) | ASTM D93 | >200 | >123 |
| Mixing ratio | - | 100 parts by weight | 30 parts by weight |
| Gel time at 30 °C | | 120 min | |
| Curing time at room temperature (25–30 °C) | | 24 h | |

$Al_2O_3$ and SiC nanoparticles developed by Sisco Research Laboratories Pvt. Ltd., Mumbai, India, were used to prepare hybrid nanocomposites. The specific properties of the nanoparticles are shown in Table 3.

**Table 3.** Typical properties of nanoparticles.

| Property | $Al_2O_3$ | SiC |
|---|---|---|
| Color | White | Grey |
| APS (nm) | 20–30 | 50 |
| Purity (%) | 99.9 | 98 |
| Melting point (°C) | 2030 | 2700 |
| Shelf life (years) | 5 | 5 |

APS—average particle size.

## 2.2. Fabrication of Neat and Hybrid Nanocomposites

The unidirectional quasi-isotropic CFRP composites (neat CFRP) were prepared with 24 layers (275 × 275 mm) in the fiber orientation of $0/-45/45/90°$ using a hand lay-up method followed by compression molding. The approximate thickness of the composite achieved was $6 \pm 0.2$ mm. The composites were subjected to room temperature curing for 24 h to prevent residual thermal stresses. A similar process was incorporated to manufacture hybrid nanocomposites by including $Al_2O_3$ and SiC at different loading conditions, as per Table 4. However, the two additional steps required to achieve uniform dispersion of nanoparticles are explained in depth in Sections 2.3 and 2.4.

**Table 4.** Composition of neat and hybrid nanocomposites.

| Composite Designation | Carbon Fiber Weight (%) | Epoxy Resin Weight (%) | Nanoparticle Weight (%) |
|---|---|---|---|
| CFRP (neat) | 50 | 50 | - |
| $Al_2O_3$ 1 wt.% | 50 | 49 | 1 |
| $Al_2O_3$ 1.5 wt.% | 50 | 48.5 | 1.5 |
| $Al_2O_3$ 1.75 wt.% | 50 | 48.25 | 1.75 |
| $Al_2O_3$ 2 wt.% | 50 | 48 | 2 |
| SiC 1 wt.% | 50 | 49 | 1 |
| SiC 1.25 wt.% | 50 | 48.75 | 1.25 |
| SiC 1.5 wt.% | 50 | 48.5 | 1.5 |
| SiC 2 wt.% | 50 | 48 | 2 |

## 2.3. Sonication and Stirring of Nanoparticles into Epoxy Resin

$Al_2O_3$ and SiC were dispersed into epoxy resin using a high-intensity probe sonicator of 2 kW with a 25 mm probe diameter. The dispersion of nanoparticles is challenging due to the dense nature of the epoxy resin because increasing the weight fraction of these nanoparticles in epoxy increases the viscosity of epoxy. Additionally, while performing the sonication process, an increase in temperature was observed, causing deterioration of the mechanical properties of the epoxy solution. To avoid this, the sonication process was performed in a pulsed mode which hinders the temperature increase rate, allowing for better temperature control [45]. For the present work, the sonication of nanoparticles was carried out in a pulsed mode with 15 s on and 30 s off, for 60 min at an amplitude setting of 50%. The detailed flow chart for the preparation of hybrid nanocomposites is shown in Figure 1.

The sonication process was further followed by mechanically stirring using a magnetic stirrer to achieve a further homogeneous distribution of nanoparticles into the epoxy solution. The epoxy solution was allowed to attain room temperature before performing the stirring operation. The nanoparticle epoxy solution was stirred at a rotational speed of 600 rev/min for 30 min.

## 2.4. Field-Emission Scanning Electron Microscope

A field-emission scanning electron microscope (FE-SEM) from Zeiss Sigma was used to investigate the uniform dispersions of the $Al_2O_3$ and SiC into epoxy resin. The specimens were cut into $10 \times 10$ mm$^2$ dimensions using an abrasive water jet cutting machine. Gold sputtering was performed on the specimens to apply a thin layer of gold, to increase the secondary electron emission and obtain a better image of the dispersion. Energy-dispersive X-ray spectrometry (EDS) was also performed to identify the composition of Al in $Al_2O_3$ and Si in SiC.

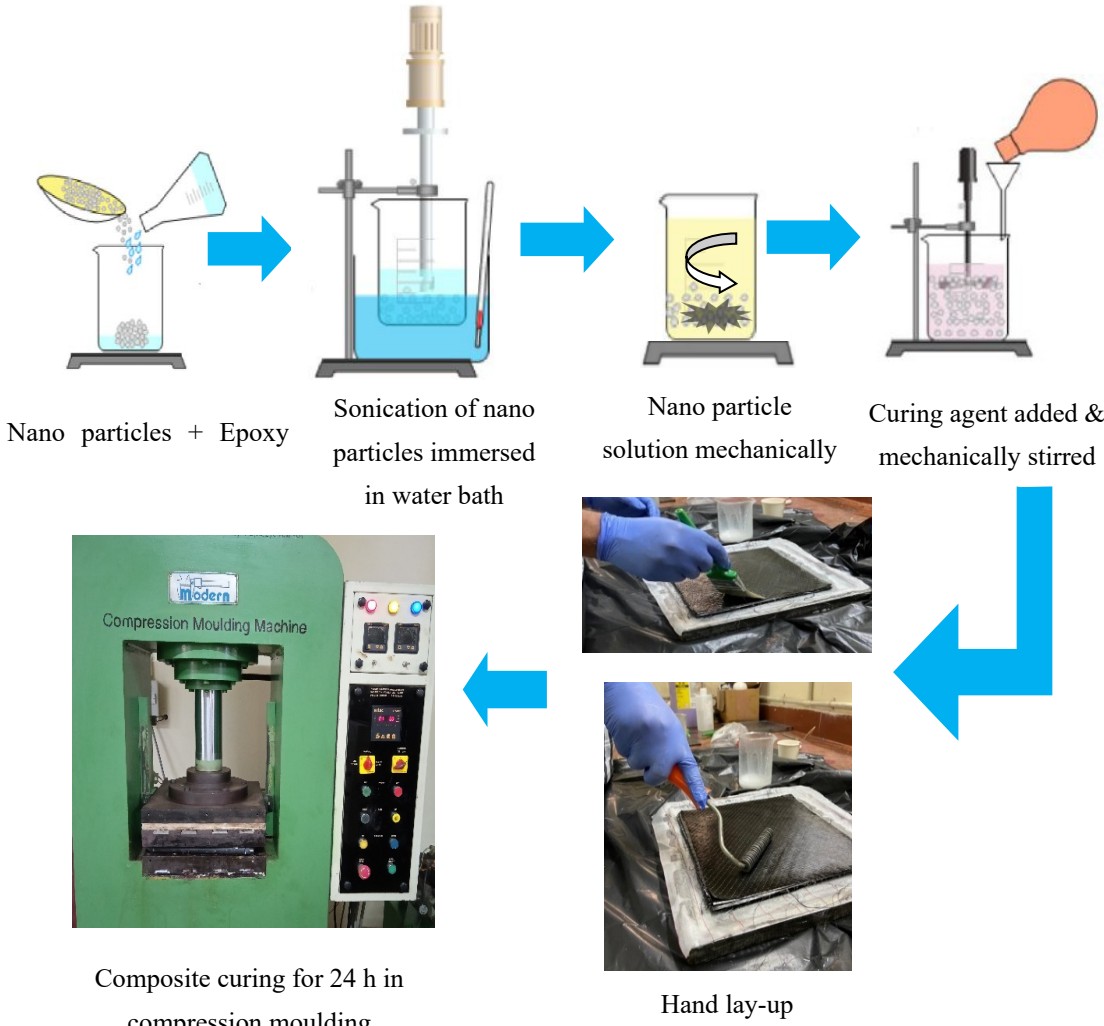

**Figure 1.** Schematic representation of fabricating the hybrid nanocomposites.

*2.5. Mechanical Characterization*

The flexural, interlaminar shear strength (ILSS), and fracture toughness properties of hybrid $Al_2O_3$ and SiC nanocomposites with various weight fractions were experimentally characterized and compared with a neat composite. A total of 5 specimens for each composition were taken and tested, and their average values were considered for determining the above properties. Dimensions of specimens were selected based on ASTM standards ASTM D7264 (flexural), ASTM D2344 (ILSS), and ISO 179 (impact), and cut using a CNC abrasive water jet cutting machine. Compared to conventional machining processes, abrasive water jet cutting is advantageous, as it eliminates the heat generated during cutting by not altering the composite's mechanical properties [46].

2.5.1. Flexural Test

The three-point bending tests were carried out on neat CFRP and hybrid $Al_2O_3$ and SiC nanocomposites following the ASTM D7264 standard procedure. The tests were performed on a Zwick/Roell Z020 testing machine with a load cell capacity of 20 kN. A crosshead speed of 1 mm/min with preload of 5 N was applied before the test was initiated. The experimental setup with the load acting on the specimen is shown in Figure 2a. The span-to-thickness ratio was maintained at 32:1. An overall length of 127 mm with a span length of 100 mm and a width of 13 mm were the dimensions of the specimens, as shown in Figure 2b. The flexural strength and modulus were determined based on Equations (1) and (2):

$$\text{Flexural strength, } \sigma_F = \frac{3P_{max}\, L}{2bh^2} \text{ (MPa)} \tag{1}$$

$$\text{Flexural modulus, } E_F = \frac{mL^3}{4bh^2} \text{ (GPa)} \tag{2}$$

where $P_{max}$—maximum load at the failure (N), b—specimen width (mm), h—specimen thickness (mm), m—initial slope at the load–deflection curve, and L—span length between two support pins (mm).

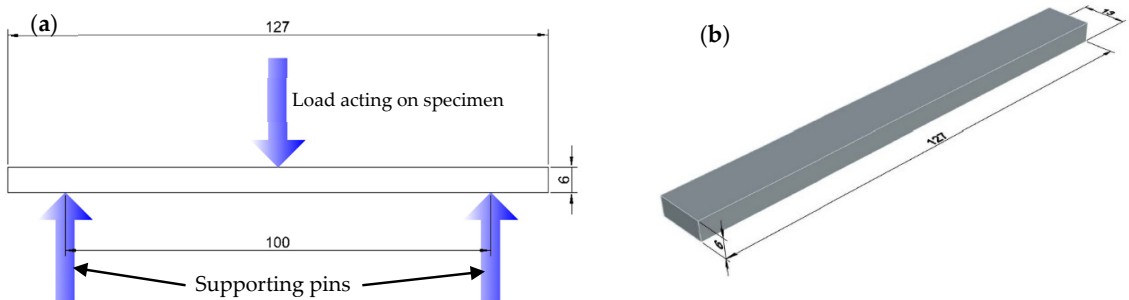

**Figure 2.** (**a**) Experimental setup of flexural test. (**b**) Isometric view of the bending specimen.

### 2.5.2. Interlaminar Shear Strength Test

The interlaminar shear strength (ILSS) of the specimens was investigated using an Instron 3366 testing machine with the crosshead speed maintained at 1 mm/min, based on ASTM D2344 standards. The experimental setup with the load acting on the specimen is shown in Figure 3a. The dimensions of the specimens were $40 \times 12 \times 6 \text{ mm}^3$ with a span-to-thickness ratio of 4, as shown in Figure 3b. Interlaminar shear strength was calculated based on Equation (3):

$$\text{ILSS, } \sigma_{sbs} = 0.75 \times \frac{P_m}{b \times h} \text{ (MPa)} \tag{3}$$

where $P_m$—maximum load at the failure (N), b—measured specimen width (mm), and h—measured thickness (mm).

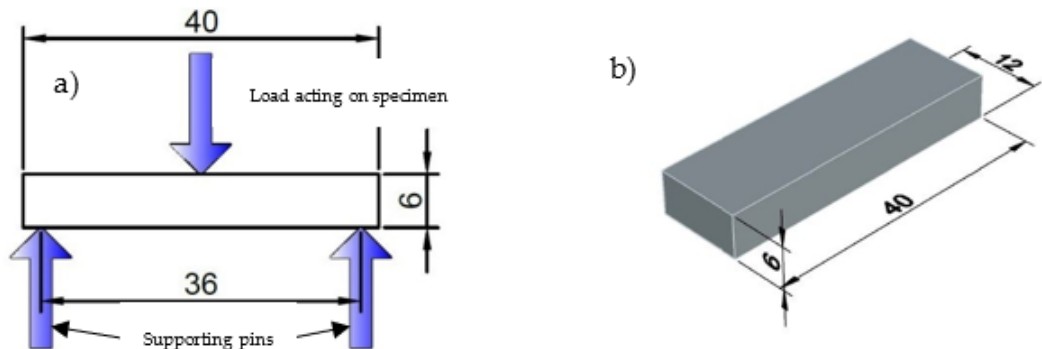

**Figure 3.** (**a**) Experimental setup of ILSS test. (**b**) Isometric view of ILSS specimen.

### 2.5.3. Impact Test

Charpy impact tests were carried out using Zwick/Roell HIT50P based on ISO 179 standards to measure the fracture toughness of neat and hybrid nanocomposites. The geometrical specifications of $80 \times 10 \times 6 \text{ mm}^3$ were selected, as shown in Figure 4. The average impact strength was calculated based on the amount of energy absorbed by the specimen before the complete fracture occurred.

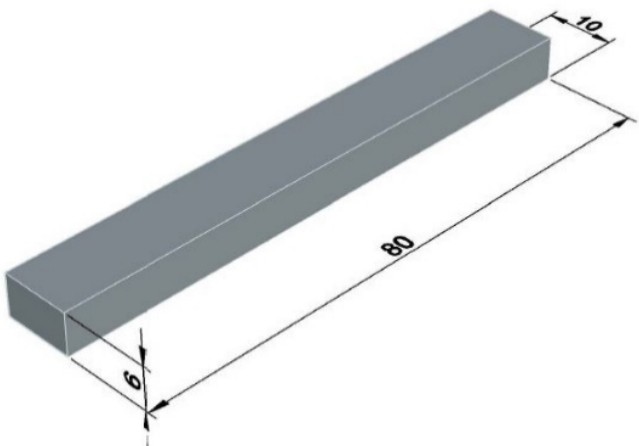

**Figure 4.** Isometric view of the Charpy specimen.

## 3. Results and Discussions

### 3.1. Morphology of Nanoparticles

Figure 5 shows a uniform distribution of nanoparticles into resin for both nanocomposites at different weight fractions. The SEM images authenticate the two-step process of sonication and magnetic stirring in effectively enhancing the dispersion of nanoparticles. As per Kaybal et al., with the increase in the weight fraction of nanoparticles, the nanoparticles agglomerated, due to the Van der Waals attractive forces, forming clusters as seen in Figure 6 [37]; thus, creating a negative effect on the mechanical properties of the nanocomposites. The detailed explanation for the mechanical properties' reduction is further reported in Sections 3.2 and 3.3.

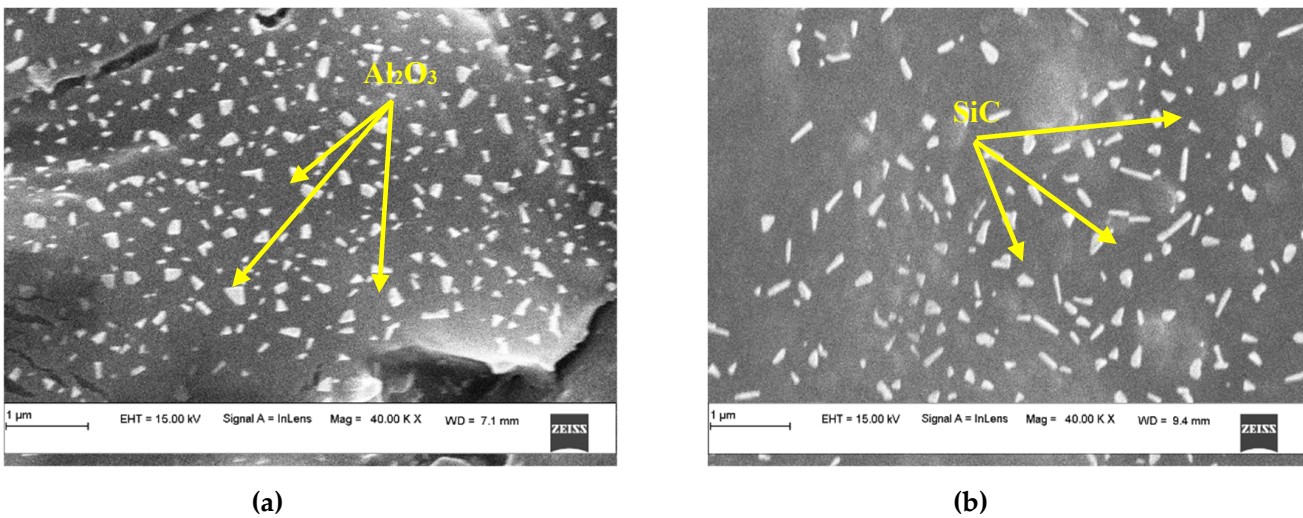

**(a)** **(b)**

**Figure 5.** Dispersion of nanoparticles: (**a**) 1.75 wt.% $Al_2O_3$, (**b**) 1.25 wt.% SiC.

EDS analysis was performed to gain the qualitative analysis of $Al_2O_3$ and SiC, and their corresponding peaks are shown in Figure 7. Table 5 offers a quantitative evaluation for Al and Si elements measured in atomic and weight %.

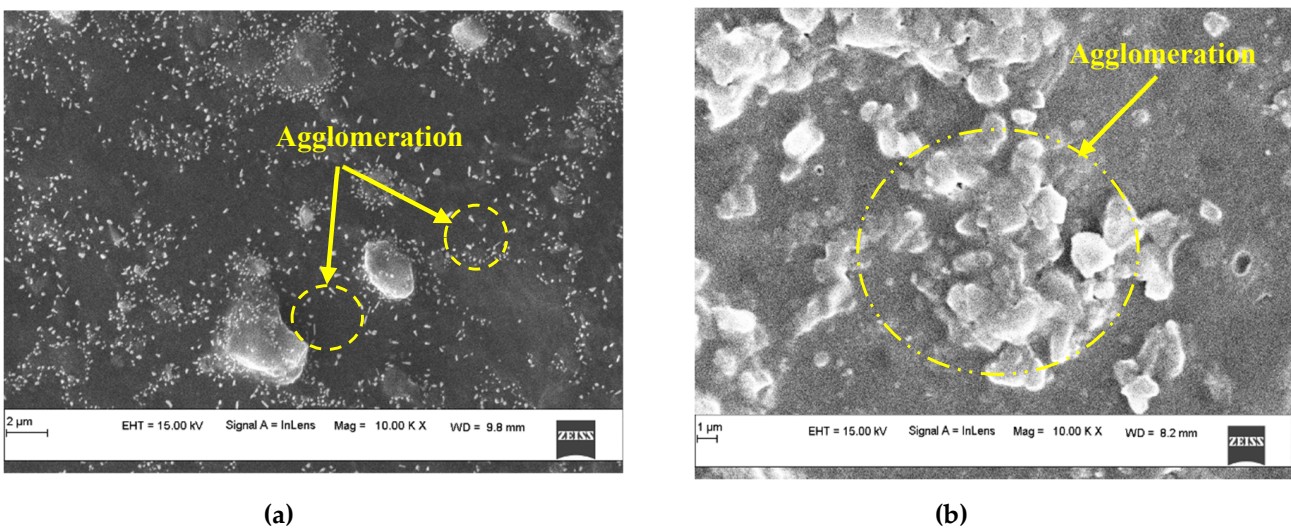

**Figure 6.** Agglomeration of nanoparticles: (**a**) 2 wt.% $Al_2O_3$, (**b**) 2 wt.% SiC.

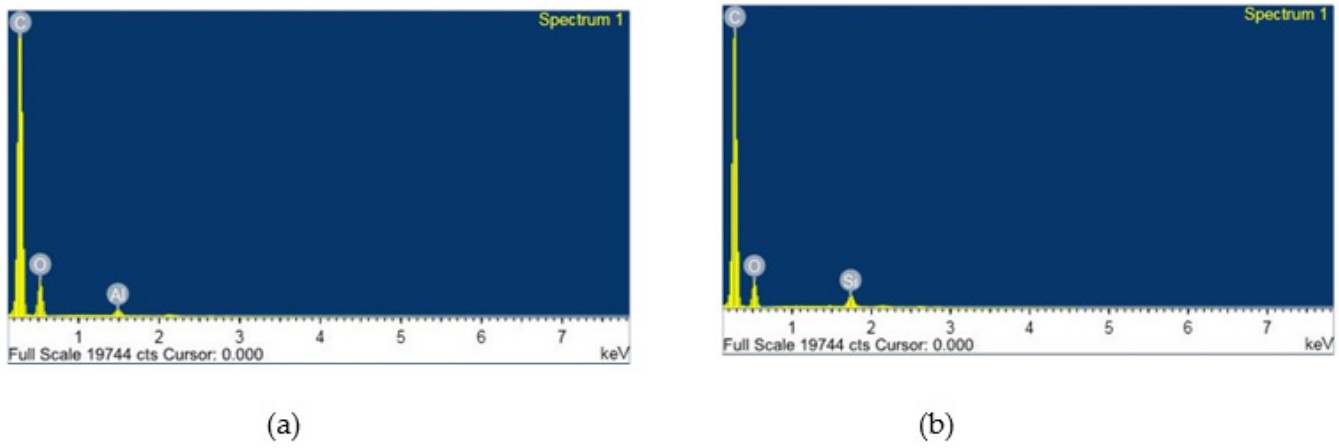

**Figure 7.** EDS graph of nanoparticles: (**a**) 1.25 wt.% $Al_2O_3$, (**b**) 1.75 wt.% SiC.

**Table 5.** EDS weight ratios of hybrid nanocomposites.

| Hybrid Nanocomposite | Alumina (Al) | | Silicon (Si) | | Carbon (C) | | Oxygen (O) | |
|---|---|---|---|---|---|---|---|---|
| | Weight % | Atomic % | Weight % | Atomic % | Weight % | Atomic % | Weight % | Atomic % |
| $Al_2O_3$ | 0.36 | 2.83 | - | - | 1.81 | 32.39 | 4.83 | 64.78 |
| SiC | - | - | 0.67 | 6.08 | 1.46 | 31.31 | 3.90 | 62.61 |

### 3.2. Flexural Behavior of Neat CFRP and Hybrid Nanocomposites

The results obtained from flexural tests are displayed in Tables 6 and 7. The results show that the flexural strength and stiffness improve with particles loaded at a specific optimal loading rate. Above this, a further addition to the matrix tends to degrade the properties, as mentioned in Section 3.1. The hybrid $Al_2O_3$ nanocomposites with 1.75 wt.% represent the maximum flexural strength and modulus, with 31.76% and 37.08% improvements over the neat composite (Table 6).

**Table 6.** Flexural behavior of neat and hybrid $Al_2O_3$ nanocomposites.

| Material | Flexural Strength (MPa) | | Strength Gain (%) | Flexural Modulus (GPa) | | Modulus Gain (%) |
|---|---|---|---|---|---|---|
| | Avg. Strength (MPa) | Std. Dev. | | Avg. Modulus (GPa) | Std. Dev. | |
| CFRP | 324.70 | 14.43 | - | 23.56 | 1.74 | - |
| $Al_2O_3$ 1 wt.% | 354.81 | 11.22 | 9.27 | 27.80 | 0.79 | 18.03 |
| $Al_2O_3$ 1.5 wt.% | 415.43 | 9.76 | 27.94 | 28.61 | 2.40 | 21.45 |
| $Al_2O_3$ 1.75 wt.% | 427.83 | 17.39 | 31.76 | 32.29 | 1.26 | 37.08 |
| $Al_2O_3$ 2 wt.% | 332.39 | 15.52 | 2.37 | 25.18 | 2.56 | 6.88 |

**Table 7.** Flexural behavior of neat and hybrid SiC nanocomposites.

| Material | Flexural Strength (MPa) | | Strength Gain (%) | Flexural Modulus (GPa) | | Modulus Gain (%) |
|---|---|---|---|---|---|---|
| | Avg. Strength (MPa) | Std. Dev. | | Avg. Modulus (GPa) | Std. Dev. | |
| CFRP | 324.70 | 14.43 | - | 23.56 | 1.74 | - |
| SiC 1 wt.% | 358.13 | 17.359 | 10.29 | 24.09 | 2.769 | 2.24 |
| SiC 1.25 wt.% | 366.25 | 15.570 | 12.79 | 25.82 | 1.475 | 9.59 |
| SiC 1.5 wt.% | 290.06 | 11.189 | −10.66 | 23.77 | 0.751 | 0.89 |
| SiC 2 wt.% | 264.22 | 13.779 | −18.62 | 22.61 | 2.359 | −4.03 |

In contrast, for the hybrid SiC nanocomposite with 1.25 wt.% filler loading, the gain obtained was relatively less, i.e., 12.79% and 9.59%, respectively (Table 7). The maximum strength and stiffness obtained for 1.75 wt.% $Al_2O_3$ were because the presence of oxygen atoms in $Al_2O_3$ when mixed with epoxy resin causes changes in epoxy chains and structures. For instance, evenly distributed and dispersed nanoparticles in the epoxy matrix decreased the epoxy chains' mobility due to the production of highly immobile nanolayers around each nanoparticle, while the matrix chains (epoxy chains that are not connected to nanoparticles) confined the non-contact matrix chains. Therefore, $Al_2O_3$ nanoparticles, which are polar particles, were thus added to create more complex network chains by filling in the spaces between the chains and thus attracting resin molecules during the curing process [47]. Additionally, the oxygen in $Al_2O_3$ nanoparticles creates effective hydrogen bonding between polymeric chains and nanoparticles, as shown in Figure 8, which tends to increase constraints between particles/polymer chains and polymer chains. These polymeric chains carry extra forces that improve the strength ability of nanocomposites compared to unfilled composites (neat) [32,48]. Figures 9 and 10 show the flexural strength and modulus variation vs. the weight fraction of $Al_2O_3$ and SiC nanoparticles.

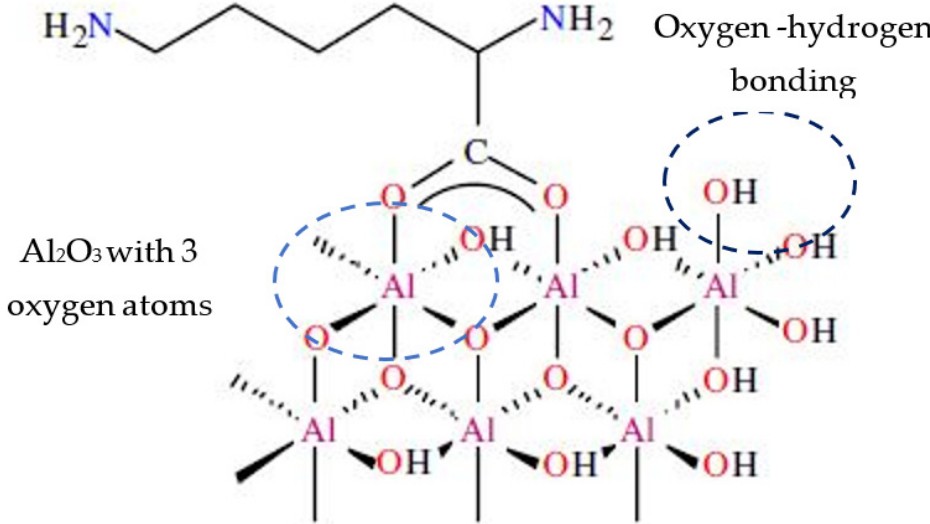

**Figure 8.** Polymeric reaction with $Al_2O_3$ nanoparticles. Reprinted with permission from [48].

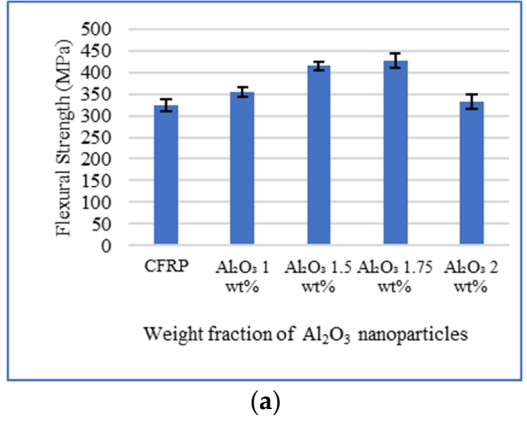 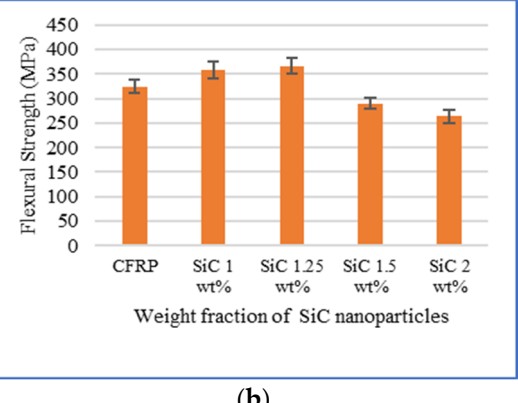

**Figure 9.** Flexural strength of neat and (**a**) hybrid $Al_2O_3$ and (**b**) hybrid SiC nanocomposites at various weight fractions.

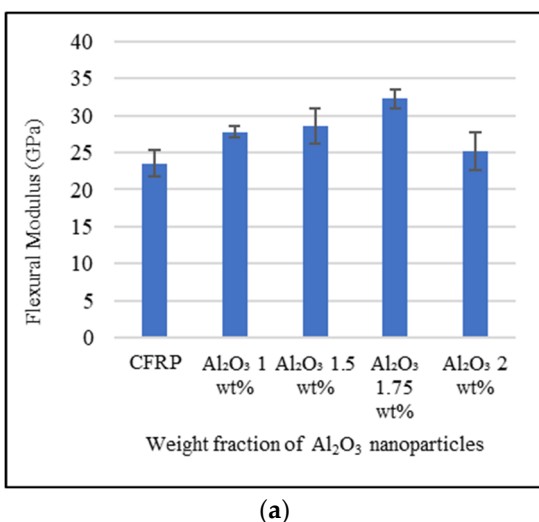 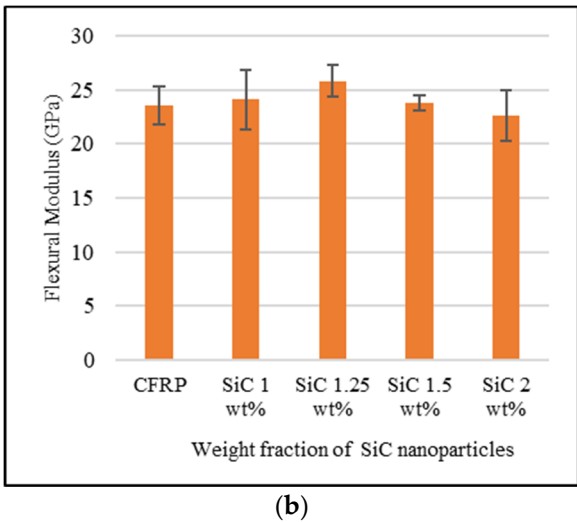

**Figure 10.** Flexural modulus of neat and (**a**) hybrid $Al_2O_3$ and (**b**) hybrid SiC nanocomposites at various weight fractions.

Similarly, in the case of the hybrid SiC nanocomposite, the optimum weight for maximum flexural strength was obtained at 1.25 wt.% filler loading. The improvement in flexural properties obtained was greater than the neat composite, but with increased filler loading (1.5 and 2 wt.%), the strength and modulus fell below the neat composite values. The explanation for the above is justified by the bonding behavior of SiC nanoparticles with epoxy resin. Similar to $Al_2O_3$ nanocomposites, as the bonding between oxygen and hydrogen takes place in the case of SiC, the bonding occurs between silica with the oxygen atom of epoxy and carbon bonds with the hydrogen atom of epoxy [49]. Due to the absence of oxygen–hydrogen bonding, which is the stronger bond compared to the carbon–hydrogen bond, during the reaction with polymer chains, the load-bearing capacity of the composite decreased. Hence, a reduction in strength was observed in comparison to $Al_2O_3$ nanocomposites. Additionally, another factor for the decline is due to the SiC nanoparticles being heavier (0.67 wt.%) particles compared to alumina (0.36 wt.%), which is lighter, as can be proven in EDS analysis. Due to this, there is a higher chance of settling of SiC particles at the bottom, creating a cluster formation that damages the flexural properties [50].

For both cases, as the amount of nanoparticles increased, strength decreased as the agglomeration took place, which caused the space (or free volume space) between polymeric chains to widen, allowing polymeric links to withstand less stress [32,51]. Various researchers noted a similar behavior of a rapid reduction in flexural strength and modu-

lus at higher filler increments of $Al_2O_3$ and SiC in epoxy-filled composites without any reinforcement fabric [52–54].

### 3.3. ILSS Behavior of Neat CFRP and Hybrid Nanocomposites

As shown in the bar graph (Figure 11), 1.75 wt.% $Al_2O_3$ can withstand loads 40.95% greater, whereas 1.25 wt.% SiC can support loads 34.27% higher than those for a neat composite. The average interlaminar shear strength and the standard deviation are presented in Tables 8 and 9. Among all the composites, the maximum ILSS was noted for the hybrid $Al_2O_3$ nanocomposite, with an average strength of 33.94 MPa. The improved shear strength was achieved due to the strong interaction between the polymer and $Al_2O_3$ particles because of Van der Waals and dipole–dipole interactions, followed by hydrogen bonding, which led to strong covalent bonding between the oxygen–hydrogen atoms [55,56]. Covalent bonding between nanoparticles and polymer lattice significantly enhances mechanical load transfer to the particles and helps promote the polymer composite's crack arresting [57]. As the filler loading exceeds the optimum value, the viscosity of epoxy increases. This further results in inadequate filler wetting and causes a reduction in shear strength [58,59].

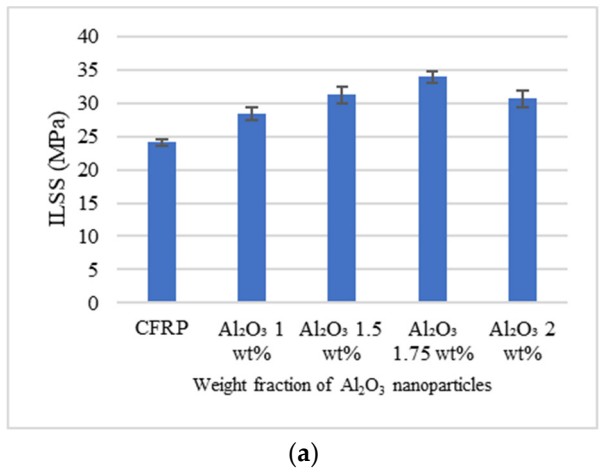

(**a**)

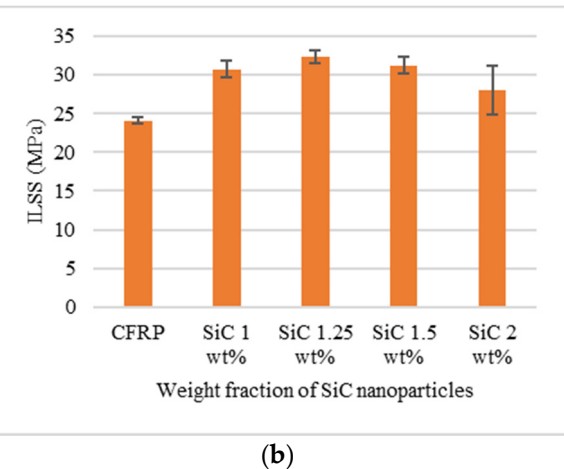

(**b**)

**Figure 11.** Interlaminar shear strength of neat and (**a**) hybrid $Al_2O_3$ and (**b**) hybrid SiC nanocomposites at various weight fractions.

**Table 8.** Interlaminar shear strength of neat and hybrid $Al_2O_3$ nanocomposites.

| Material | Interlaminar Shear Strength (MPa) | | Strength Gain (%) |
|---|---|---|---|
| | Avg. Strength (MPa) | Std. Dev. | |
| CFRP | 24.08 | 0.44 | - |
| $Al_2O_3$ 1 wt.% | 28.42 | 0.89 | 18.02 |
| $Al_2O_3$ 1.5 wt.% | 31.26 | 1.28 | 29.84 |
| $Al_2O_3$ 1.75 wt.% | 33.94 | 0.92 | 40.95 |
| $Al_2O_3$ 2 wt.% | 30.70 | 1.24 | 27.51 |

**Table 9.** Interlaminar shear strength of neat and hybrid SiC nanocomposites.

| Material | Interlaminar Shear Strength (MPa) | | Strength Gain (%) |
|---|---|---|---|
| | Avg. Strength (MPa) | Std. Dev. | |
| CFRP | 24.08 | 0.44 | - |
| SiC 1 wt.% | 30.74 | 1.07 | 27.67 |
| SiC 1.25 wt.% | 32.33 | 0.87 | 34.27 |
| SiC 1.5 wt.% | 31.22 | 1.04 | 29.69 |
| SiC 2 wt.% | 28.00 | 3.10 | 16.30 |

The SEM images of the ILSS specimen with an increasing magnification factor, after failure, with various damage characteristics are presented in Figure 12. It is evident from the SEM images that the maximum cracks were developed for the neat composite (Figure 12a). The intensity of crack propagating was higher (i.e., along the composite length) as no nanoparticles were embedded into it. However, in the case of hybrid nanocomposites (Figure 12 b,c), the cracks formed along the length were fewer, as there was a resistance offered by the nanoparticles acting as crack arrestors, which overall increased the shear strength of the composites.

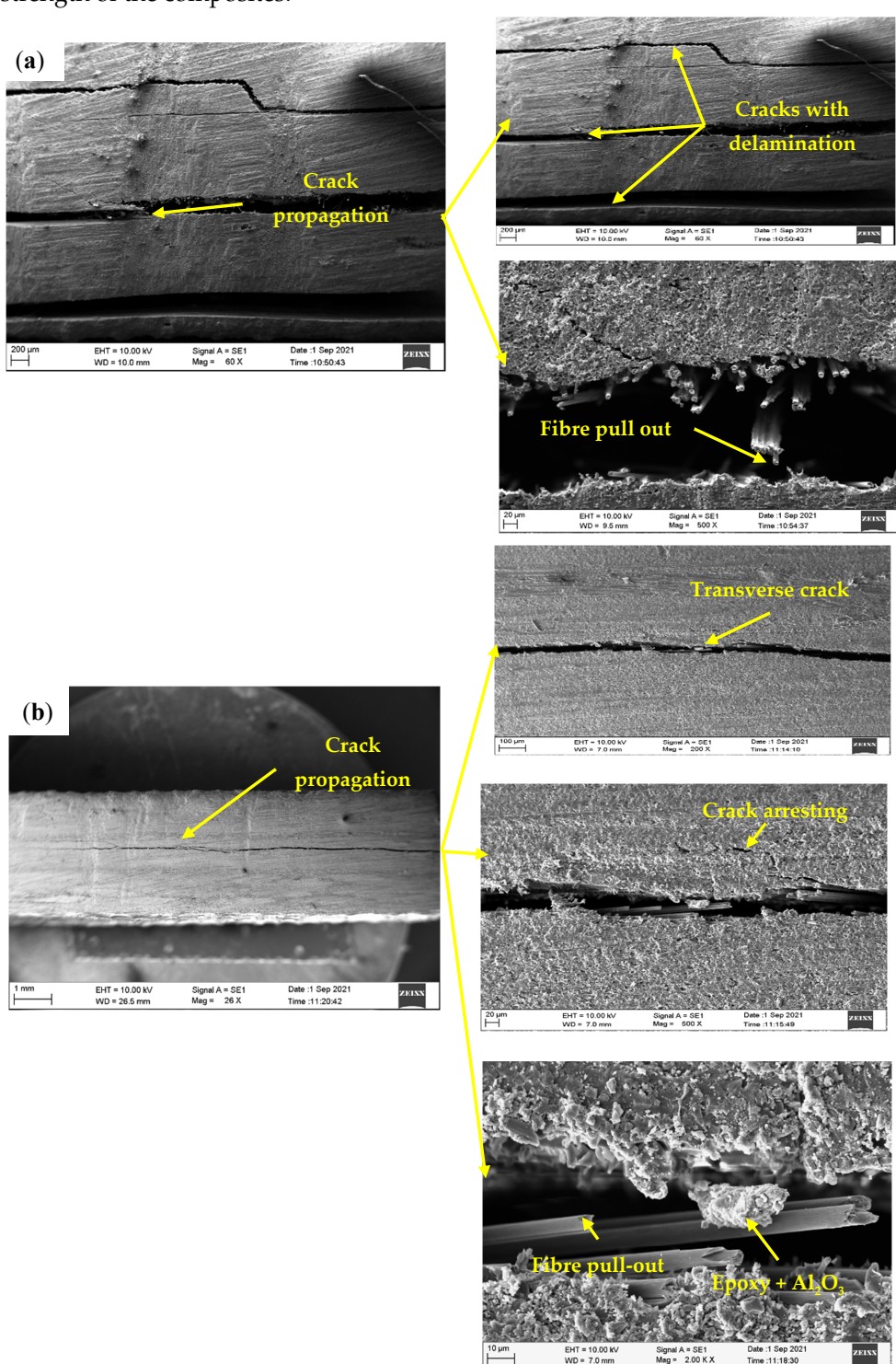

**Figure 12.** *Cont.*

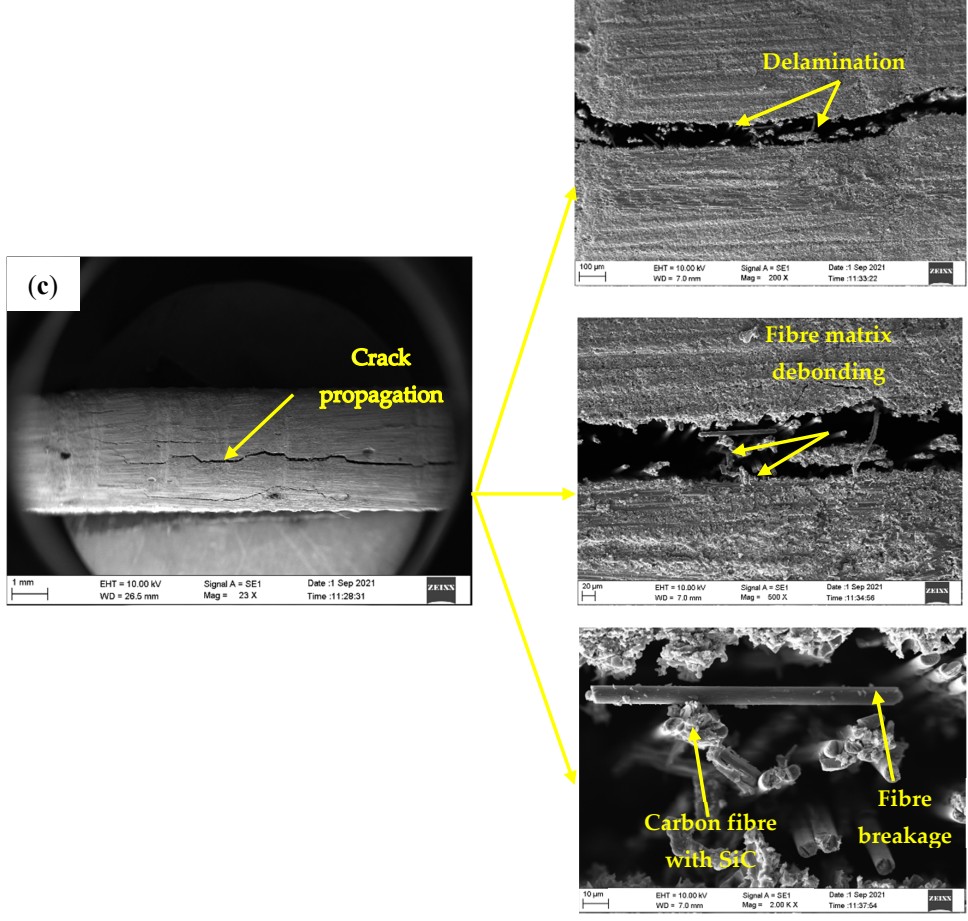

**Figure 12.** SEM images of shear test fracture specimens: (**a**) neat, (**b**) 1.75 wt.% $Al_2O_3$, and (**c**) 1.25 wt.% SiC nanocomposites at different magnification factors.

### 3.4. Impact Behavior of Neat CFRP and Hybrid Nanocomposites

A Charpy impact test was performed to evaluate the fracture toughness of neat and hybrid nanocomposites. Figure 13 presents the bar graph of the impact strength for different weight percentages of nanoparticles. It is observed from Tables 10 and 11 that the impact strength sharply increased for the hybrid $Al_2O_3$ nanocomposites relative to that of the hybrid SiC nanocomposites over neat composites. Fracture toughness increased by 39.52%, 43.06%, and 47.51% for 1, 1.5, and 1.75 wt.% filler loading in the case of the hybrid $Al_2O_3$ nanocomposites. Similarly, for the hybrid SiC nanocomposites, improvement was observed at an increment of 17.16% and 30.45% for 1 and 1.25 wt.% filler loading, respectively. The factors that signify the improvement in toughness for both the hybrid nanocomposites are: (a) the uniform distribution of nanoparticles offered improved resistance to the propagating crack, and (b) a decreased crack length was subjugated as a result of crack deflection taking place. Crack deflection and crack arresting absorb and consume more energy at the crack front, delaying crack extension that further enhancing the nanocomposites' fracture toughness [60,61]. The reduction in the toughness property at higher filler loading (above 1.75 wt.% for $Al_2O_3$ and 1.25wt.% for SiC) was produced due to the high particle–particle interaction causing agglomeration of nanoparticles and lower resistance to the crack propagation taking place during failure. Due to agglomeration, the polymer failed to penetrate between the filler particles, culminating in a loose bond with the epoxy. When the crack propagation interacted with clustered particles, the loosely bonded particles disassociated from the matrix, forming voids, resulting in the complete separation of particles. This crack propagation phenomenon in nanocomposites absorbs less energy without providing any resistance to the crack during failure [62–64]. As a result, the

fracture toughness decreased with the increasing filler content after an optimum level. A similar trend in the behavior of increments in fracture toughness and the rapid reduction in the load-absorbing capacity of $Al_2O_3$ and SiC filled with increased filler loading was reported by Karapappas et al. [65] and Kychkin et al. [66].

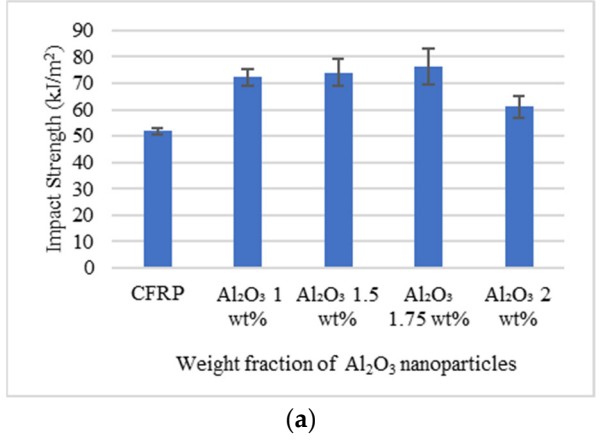

(**a**)

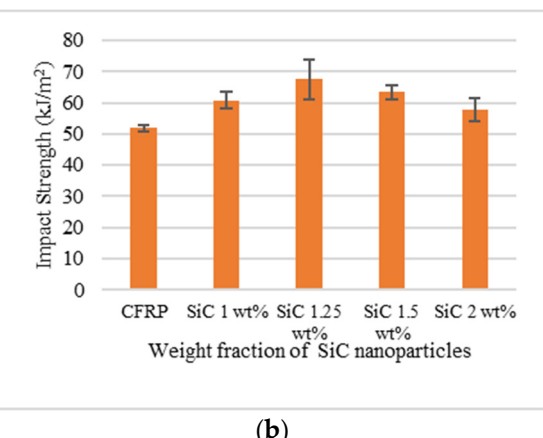

(**b**)

**Figure 13.** Impact strength of neat and (**a**) hybrid $Al_2O_3$ and (**b**) hybrid SiC nanocomposites at various weight fractions.

**Table 10.** Impact strength of neat and hybrid $Al_2O_3$ nanocomposites.

| Material | Impact Strength (kJ/m$^2$) | Standard Deviation | Strength Gain (%) |
|---|---|---|---|
| CFRP | 51.81 | 1.13 | - |
| $Al_2O_3$ 1 wt.% | 72.29 | 2.31 | 39.52 |
| $Al_2O_3$ 1.5 wt.% | 74.12 | 6.39 | 43.06 |
| $Al_2O_3$ 1.75 wt.% | 76.43 | 3.10 | 47.51 |
| $Al_2O_3$ 2 wt.% | 61.18 | 4.07 | 18.08 |

**Table 11.** Impact strength of neat and hybrid SiC nanocomposites.

| Material | Impact Strength (kJ/m$^2$) | Standard Deviation | Strength Gain (%) |
|---|---|---|---|
| CFRP | 51.814 | 1.133 | - |
| SiC 1 wt.% | 60.706 | 2.650 | 17.16 |
| SiC 1.25 wt.% | 67.595 | 6.392 | 30.45 |
| SiC 1.5 wt.% | 63.401 | 2.312 | 22.36 |
| SiC 2 wt.% | 57.797 | 3.707 | 11.54 |

## 4. Conclusions

The present experimental study explained the role of $Al_2O_3$ and SiC nanoparticles in enhancing the mechanical properties of carbon fiber-reinforced polymer composites. The major outcome of this work was achieving an optimum loading condition of the above nanoparticles. The following conclusions can be drawn from the study.

1. The maximum flexural, shear, and impact strength improvements were obtained at 1.75 wt.% $Al_2O_3$ and 1.25 wt.% SiC nanoparticles' loading over neat composites.
2. The mechanical properties were enhanced by the proper selection of the ultrasonication parameters and the combination of magnetic stirring methods that enabled the effective dispersion of nanoparticles.
3. Higher filler loading above the optimum level (i.e., 2 wt.% for $Al_2O_3$ and 1.5 wt.% for SiC) reduced the mechanical properties of hybrid nanocomposites.

4   Flexural strength and modulus were seen as maximum for hybrid $Al_2O_3$ nanocomposites. In contrast, a significant drop was observed for the hybrid SiC nanocomposites, above the optimum level of nanoparticles' loading, falling below the strength value of neat composites.

5   $Al_2O_3$ nanocomposites were more effective in improving the properties than SiC nanocomposites due to the strong covalent bond formation of the particles' interaction with polymeric chains of epoxies.

**Author Contributions:** Investigation, methodology, writing—original draft preparation, S.M.S.; conceptualization, writing—review and editing, P.K.S.; conceptualization, methodology, writing—review and editing, supervision, validation, N.S.; conceptualization, methodology, writing—review and editing, project administration, S.S.; supervision, project administration, S.D.S.; supervision, project administration, N.N. All authors have read and agreed to the published version of the manuscript.

**Funding:** This research received no external funding.

**Data Availability Statement:** Not applicable.

**Conflicts of Interest:** The authors declare no conflict of interest.

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
