# Peer review of "Effect of Alumina and Silicon Carbide Nanoparticle-Infused Polymer Matrix on Mechanical Properties of Unidirectional Carbon Fiber-Reinforced Polymer"

_jcs, doi:10.3390/jcs6120381_

Round 1

Reviewer 1 Report

The results and presentation are commendable, however, a proof reading and editing by a professional would be advised.  The paper can be accepted for publication subject to the following minor corrections.

1. The literature review is extensive, though some part of it is not directly related to the results presented. Authors are requested to reduce the literature review to be concise and keep it as connected to the body of the science presented in the paper. 

2.  Experimental results section: It is not required in a technical discussion to provide minute details like manufacturer of the resin etc. In Table 2, avoid adding visual inspection as a legitimate reportable observation. 

3. Sections 2.3 and 2.4 can be combined. To allow the reader to understand and better appreciate the scientific merit, it is advised to remove trivial information. Authors are gently reminded that the reader should remain entertained to appreciate the results presented, thus, lengthy explanations of trivial information might divert the attention of the reader. 

4. Explanation of 2.6.1 should be supplemented with a drawing of the specimen, showing the dimensions. A preload given in the testing machine should be reported as a load (in N) as opposed to a stress (0.1MPa?).

5. Figure 2 lacks any context and explanation. Please add more info on the figure label.

6. Further, this images 2-5 will not meet the standard to publish in a peer reviewed journal. Better images are expected. 

7. Please keep uniformity in the images 9-11 and 13. 

Author Response

Response (revision notes) to the Reviewer’s Comments:

The authors of this paper would like to sincerely thank the reviewers for their valuable comments/suggestions made. Please find the response for each comment in the following paragraphs.

  1. The literature review is extensive, though some part of it is not directly related to the results presented. Authors are requested to reduce the literature review to be concise and keep it as connected to the body of the science presented in the paper. 

Response: As per the reviewer’s suggestion the literature part which is not directly related to the topic has been removed in the Section 1. Introduction.

  1. Experimental results section: It is not required in a technical discussion to provide minute details like manufacturer of the resin etc. In Table 2, avoid adding visual inspection as a legitimate reportable observation. 

Response: As per the reviewer’s suggestion, the manufacturers details has been removed in Section 2.1. Also, From Table 2, the visual inspection part has been removed.

  1. Sections 2.3 and 2.4 can be combined. To allow the reader to understand and better appreciate the scientific merit, it is advised to remove trivial information. Authors are gently reminded that the reader should remain entertained to appreciate the results presented, thus, lengthy explanations of trivial information might divert the attention of the reader. 

Response: Section 2.3 and 2.4 has been combined in the revised manuscript. As per the reviewer’s suggestion the lengthy explanation which diverts the topic has been removed in the Section 2.3.

  1. Explanation of 2.6.1 should be supplemented with a drawing of the specimen, showing the dimensions. A preload given in the testing machine should be reported as a load (in N) as opposed to a stress (0.1MPa?).

Response: In the revised manuscript, in section 2.5.1, the old specimen figures have been replaced with the experimental setup and isometric view of different specimens with specific dimensions (i.e., Figure 2-4). Also, the preload given is mentioned as 5 N.

  1. Figure 2 lacks any context and explanation. Please add more info on the figure label.

Response: Figure 2 has been replaced with experimental setup and isometric view image.

  1. Further, this images 2-5 will not meet the standard to publish in a peer reviewed journal. Better images are expected. 

Response: Figures 2-5, have been replaced with high quality images in the revised manuscript.

  1. Please keep uniformity in the images 9-11 and 13. 

Response: We respect and acknowledge the reviewer’s suggestion, but as Figure 12 is a part of explanation of Section 3.3. We need to keep the content as it is in the same format. We request you to consider our decision.

Reviewer 2 Report

Line 31: CFRP could use a more detailed explanation to readers who are not familiar with the concept.

Figure 8: What are some of the experimental evidences that prove that the nanoparticle is bonding with resin?

Line 340: Is particle wetting and covalent bond forming the same concept here or not?

Author Response

Response (revision notes) to the Reviewer’s Comments:

The authors of this paper would like to sincerely thank the reviewers for their valuable comments/suggestions made. Please find the response for each comment in the following paragraphs.

Line 31: CFRP could use a more detailed explanation to readers who are not familiar with the concept.

Response: As per the reviewer’s suggestion a detailed explanation regarding the CFRP has been added in the revised manuscript in Section 1. Introduction.

Figure 8: What are some of the experimental evidences that prove that the nanoparticle is bonding with resin?

Response: As per the authors knowledge from literature performed, Energy dispersive spectroscopy (EDS) and Fourier transform infrared spectroscopy (FTIR) are the two experimental investigation that proves the bonding of nano particles with resin.

Line 340: Is particle wetting and covalent bond forming the same concept here or not?

Response: Yes, in the present investigation authors have considered particle wetting and covalent bond forming to be the same concept.

Round 2

Reviewer 1 Report

Thank you for your quick response.

The comment 7 in my review meant to keep uniformity in the images 9-11 and 13-- this means, use the same color for plots, bounding box (might even remove the bounding boxes in blue and grey) and make sure subfigures (a) and (b) are aligned (For example, Figure 13(a) and 13(b) are not aligned properly. Such minute but visible details should in general be addressed while submitting the paper to a  reputed journal).  All the other edits are sound and I will recommend the paper be accepted for publication. 

Reviewer 2 Report

The reviewer would like to thank the authors for properly addressing the reviewer's concern. The reviewer recommends this paper be accepted.